# Environmental Endocrinology: Parabens Hazardous Effects on Hypothalamic–Pituitary–Thyroid Axis

**DOI:** 10.3390/ijms242015246

**Published:** 2023-10-17

**Authors:** Damáris Barcelos Cunha Azeredo, Denilson de Sousa Anselmo, Paula Soares, Jones Bernardes Graceli, D’Angelo Carlo Magliano, Leandro Miranda-Alves

**Affiliations:** 1Laboratory of Experimental Endocrinology-LEEx, Institute of Biomedical Sciences, Federal University of Rio de Janeiro, Rio de Janeiro 21941-902, Brazil; damaris.barceloscunha@gmail.com (D.B.C.A.); denilson_anselmo@hotmail.com (D.d.S.A.); dc.magliano@gmail.com (D.C.M.); 2Postgraduate Program in Endocrinology, Faculty of Medicine, Federal University of Rio de Janeiro, Rio de Janeiro 21941-902, Brazil; 3Cellular Signaling and Metabolism Group, i3S—Institute for Research and Innovation in Health, University of Porto, 420-135 Porto, Portugal; psoares@ipatimup.pt; 4Department of Pathology, Faculty of Medicine, University of Porto, 4200-139 Porto, Portugal; 5Laboratory of Cellular Toxicology and Endocrinology, Department of Morphology, Federal University of Espírito Santo, Vitória 29047-105, Brazil; jbgraceli@gmail.com; 6Morphology and Metabolism Group, Federal University of Fluminense, Niteroi 24020-150, Brazil; 7Postgraduate Program in Pharmacology and Medicinal Chemistry, Institute of Biomedical Sciences, Federal University of Rio de Janeiro, Rio de Janeiro 21941-902, Brazil; 8Postgraduate Program in Morphological Sciences, Institute of Biomedical Sciences, Federal University of Rio de Janeiro, Rio de Janeiro 21941-902, Brazil

**Keywords:** parabens, toxicity, thyroid, endocrine disruptor, hypothalamus–pituitary–thyroid axis

## Abstract

Parabens are classified as endocrine-disrupting chemicals (EDCs) capable of interfering with the normal functioning of the thyroid, affecting the proper regulation of the biosynthesis of thyroid hormones (THs), which is controlled by the hypothalamic–pituitary–thyroid axis (HPT). Given the crucial role of these hormones in health and the growing evidence of diseases related to thyroid dysfunction, this review looks at the effects of paraben exposure on the thyroid. In this study, we considered research carried out in vitro and in vivo and epidemiological studies published between 1951 and 2023, which demonstrated an association between exposure to parabens and dysfunctions of the HPT axis. In humans, exposure to parabens increases thyroid-stimulating hormone (TSH) levels, while exposure decreases TSH levels in rodents. The effects on THs levels are also poorly described, as well as peripheral metabolism. Regardless, recent studies have shown different actions between different subtypes of parabens on the HPT axis, which allows us to speculate that the mechanism of action of these parabens is different. Furthermore, studies of exposure to parabens are more evident in women than in men. Therefore, future studies are needed to clarify the effects of exposure to parabens and their mechanisms of action on this axis.

## 1. Introduction

The incidence of thyroid dysfunction (TD) has increased worldwide recently, particularly among women. Data from the literature show an increase of TD in women during reproductive age in a ratio of 4:1 compared to men. Furthermore, thyroid cancer has a higher incidence in women than in men [1,2,3,4], which suggests some influence of hormonal regulation on the development and establishment of TD. Nevertheless, men can also be affected by TD.

The HPT axis is a major modulator of synthesis and regulation of the prohormone tetraiodothyronine (T4) and the active hormone triiodothyronine (T3), both of which are crucial for the body’s normal growth, development, and homeostasis. In the HPT axis, the thyrotropin-releasing hormone (TRH) is synthesized and secreted by the hypothalamus. TRH acts on its receptor in the pituitary, stimulating thyroid-stimulating hormone (TSH), which acts in the thyroid gland through its receptors, stimulating the production of the THs (T3 and T4). The THs are responsible for a negative feedback loop that regulates the secretion of TRH and TSH, reducing THs production and promoting hormone balance in the body [5,6,7,8].

The mechanism of THs synthesis and secretion is very complex and highly regulated and is influenced by endogenous and exogenous factors. Within this context, environmental factors, such as nutrients, viruses, and radiation, are determinants for the appearance of TD, but also the environmental pollution caused by EDCs needs to be considered. Many studies have reported a relation between EDCs exposure and alterations in the HPT axis function [9,10,11,12,13,14,15,16]. EDC compounds have been considered one of the major factors associated with functional disruption of the thyroid [17,18,19,20,21,22].

According to the Environmental Protection Agency (EPA) U.S., EDCs are any chemicals that can interfere with the normal functions of the endocrine system and lead to problems with reproduction (e.g., egg and sperm production) and development (e.g., healthy fetal growth) in both humans and wildlife [23]. Conversely, according to the World Health Organization (WHO), EDCs are exogenous substances or mixtures that alter the function(s) of the endocrine system and consequently cause adverse health effects in an intact organism, its progeny, or (sub)populations [24].

The EDCs can act by different mechanisms, namely: (1) interacting with or activating hormone receptors; (2) antagonizing hormone receptors; (3) altering hormone receptor gene and protein expression; (4) altering signal transduction in hormone-responsive cells; (5) inducing epigenetic modifications in hormone-producing or hormone-responsive cells; (6) altering hormone synthesis; (7) altering hormone transport across cell membranes; (8) altering hormone distribution or circulating levels of hormones; (9) altering hormone metabolism or clearance; and (10) altering the fate of hormone-producing or hormone-responsive cells [25]. However, the mechanisms of action of EDCs depend on specific actions at the cellular and tissue levels, as well as on circadian rhythms, seasonal changes, life stage, and sex [25].

Finally, EDCs can be classified according to their origin (natural or synthetic) and grouped according to their chemical composition. Compounds that are excreted by living beings are considered natural, such as phytoestrogens, flavonoids, and natural estrogens. Synthetic compounds can be of industrial or domestic origin, including products such as bisphenols (e.g., plastics), phthalates (e.g., plasticizers), heavy metals, pesticides, retardants (e.g., computers), and parabens (e.g., cosmetics). The main routes of exposure to these compounds are dermal, diet, or inhalation [26,27,28,29].

An increasing number of studies have been published on the association between exposure to parabens and endocrine-related diseases, especially in susceptible people, such as pregnant women and children. However, the physiological mechanisms involved in exposure to these compounds are still not fully understood, as there are few studies in the scientific literature that demonstrate the impacts of exposure to these compounds on the health of organisms. Therefore, we propose to review the evidence in the literature correlating exposure to parabens and the development of TD with a focus on human and animal models.

## 2. Thyroid Morphophysiology: An Overview

In mammals, the thyroid gland is composed of two lobes (right and left) situated anterolaterally to the trachea. The thyroid tissue is composed of several follicles and a large amount of blood vessels. The blood vessels in the thyroid are responsible for transporting oxygen and nutrients to thyroid cells, allowing them to perform their metabolic functions and produce thyroid hormones in adequate quantities. Additionally, these blood vessels also help remove waste products and metabolites from thyroid tissue. Finally, the hormones produced in the gland are released into the bloodstream to be distributed throughout the body. In summary, the blood vessels in the thyroid play a vital role in supporting the thyroid gland’s function, ensuring the proper distribution and production of thyroid hormones. To a lesser extent, the parafollicular C cells produce calcitonin, which, together with the parathyroid hormone produced in the parathyroid, acts on calcium metabolism [30,31,32]. The main functional part of this gland is the thyroid follicle, which is widely distributed along the thyroid and is supported by loose connective tissue. These follicles are oval-shaped structures with a three-dimensional configuration, and their lumen is composed of colloid, a gelatinous substance composed of iodinated and non-iodinated thyroglobulin (TG), diiodothyronine (DIT), monoiodothyronine (MIT), T3, and T4. The lining of the follicle is composed of epithelial cells called follicular cells and thyrocytes, which can be cuboidal or squamous depending on the pituitary stimulus to produce the THs [33,34,35,36].

The main compound of the colloid is TG, a glycoprotein synthesized by follicular cells with tyrosine residues in its composition. TSH, by binding to the TSH receptor (TSHr) located in the basal domain, stimulates the production of several proteins involved in the synthesis of THs, such as TG. This precursor protein of THs has a sequence dominated by several cysteine-rich domains, a molecular weight equivalent to 600 kDa, and remarkable stability and solubility due to many disulfide bridges per monomer and about seventeen glycosylation sites. After its synthesis in thyrocytes, TG is secreted into the colloid, where it is stored. The synthesis and secretion of THs are dependent on iodine, and thus, TSH stimulates iodine uptake against the concentration gradient, increasing ion concentrations in the cell cytoplasm and in the follicle lumen. Thus, in the apical membrane of follicular cells, tyrosine residues are iodinated to iodotyrosine by thyroid peroxidase (TPO), an enzyme that produces THs in a reaction dependent on hydrogen peroxide, produced by the enzyme DUOX (Dual oxidase), which is also present in this region of the cell. After this step, a portion of the colloid undergoes endocytosis by thyrocytes and digestion by the action of cytoplasmic lysosomes. The TG is proteolyzed and releases free TH into the cytoplasm which will later be directed to the bloodstream. The remaining iodide from this reaction is recycled by the action of thyroid dehalogenase (Dehal1) and used again for hormone biosynthesis [37,38,39,40,41,42].

Hydrogen peroxide production is an essential step for iodide oxidation and thyroglobulin iodination for THs biosynthesis. In the thyroid, the oxidases Dual oxidase 1 (DUOX1) and Dual oxidase 2 (DUOX2) stand out, which are members of the NADPH oxidase (NOX) family of oxidoreductase enzymes, which are dependent on calcium to generate hydrogen peroxide. Under normal conditions, dual oxidases are highly expressed in the thyroid and other tissues (salivary gland, gastrointestinal tract). However, DUOX1 and DUOX2 are expressed only under physiological conditions in the thyroid [43]. There is 83% sequence similarity between DUOX1 and DUOX2, but they are differently regulated via direct phosphorylation: DUOX1 is activated by protein kinase A and DUOX2 is activated by protein kinase C. Both pathways are activated by calcium [44,45,46].

It is well established in the literature that TPO uses the hydrogen peroxide produced by DUOX to promote the oxidation of dietary iodine, which is then captured with the aid of the sodium/iodide symporter (NIS). This oxidized iodine is coupled to the tyrosine residues present in thyroglobulin, thus promoting the synthesis of THs. Although hydrogen peroxide is crucial for the biosynthesis of THs, when found in high concentrations of reactive oxygen species (ROS) in the body, it can have adverse effects on health. ROS include the superoxide anion (O_2_^−^), hydrogen peroxide, and hydroxyl radicals (OH), among others [47,48].

ROS formation can occur through enzymatic or non-enzymatic reactions when there is an imbalance between prooxidant factors and their elimination. This can be related to the action of endogenous or exogenous factors and can lead to a range of molecular damage [49,50,51,52,53,54,55]. The maintenance of redox homeostasis is promoted by molecules with antioxidant potential, which act in the regulation of ROS production and elimination. For this reason, the thyroid has a highly complex antioxidant system to protect its integrity, as it is continuously exposed to ROS for normal function and THs biosynthesis [56]. Some enzymes reduce ROS by minimizing or delaying the effects caused by these reactive species, providing a primary antioxidant defense, such as catalase, superoxide dismutase, glutathione peroxidase, and glutathione reductase [54,55,56,57,58,59].

## 3. Parabens

Parabens are chemical compounds classified as alkyl esters of parahydroxybenzoic acid (PHBA). Common parabens include benzylparaben (BeP), butylparaben (BuP), ethylparaben (EP), methylparaben (MP), and propylparaben (PP), which have structural differences between them (Table 1). These compounds show antifungal and antimicrobial potential and are widely used as preservatives in food, beverages, drugs, papers, and personal care products [60,61]. The main source of exposure is dermal absorption, but ingestion of products containing parabens is also an important pathway of exposure in the general population [62,63,64]. After ingestion, parabens are absorbed in the gastrointestinal tract and hydrolyzed by intestinal and liver esterases. The main metabolite is parahydroxybenzoic acid (PHBA), which is excreted as p-hydroxyhippuric acid (PHHA) in urine, bile, and feces within 24 to 48 h, making urinary paraben concentrations used as long-term urinary biomarkers in studies investigating human exposure levels [65,66,67,68,69,70,71].

The indiscriminate use of these compounds in various products has aroused scientific interest in the effects of exposure to parabens. Since then, it has been shown that parabens have a potential estrogenic action capable of interfering with the body’s homeostasis and thus could be classified as an EDC [72,73,74]. To regulate the use of parabens in Brazil, the Agência Nacional de Vigilância Sanitaria (ANVISA, Brazil) allows the use of up to 0.4% of individual parabens and up to 0.8% of conjugated parabens in personal care products, but there are no restrictions on the use of parabens in food products, except for propylparaben, which was banned [75]. In the European Union (EU), the maximum permissible use concentration is 0.4% for MP or EP and 0.19% for BuP or PP. As for its use in food, the European Food Safety Authority (EFSA) determines that the acceptable daily intake concentration is up to 10 mg/kg/day for MP or BuP [76,77].

In 2004, the EFSA review panel determined the No Observed Adverse Effect Level (NOAEL) for MP and EP to be 1000 mg/kg/day but considered that more data were needed to determine a specific NOAEL value for propylparaben [76]. Later, in 2008, the Cosmetic Ingredient Review (CIR) Expert Panel reviewed the safety assessment of MP, EP, PP, IPP, BuP, IBP, and BeP in cosmetic products, where it was determined that the NOAEL was 1000 mg /kg/day based on the results of Hoberman et al. (2008), which was considered the “most statistically powerful and well-conducted study on the effects of butylparaben on the male reproductive system” [77,78,79].

Despite this, several studies have demonstrated the harmful effects of exposure to parabens on the general population’s health, even at concentrations considered safe by ANVISA (Brazilian Health Regulatory Agency), raising a series of concerns. Several parabens have also been found in human biological samples [80,81,82,83,84,85]. In samples of blood and breast milk, concentrations of 0.62 ng/mL of MP, 1.03 ng/mL of EP, 0.18 ng/mL of PP, and 0.05 ng/mL of BuP were found [82]. These compounds were also found in placental tissue samples, with concentrations of up to 11.77 ng/g of MP, EP, and PP [83]. In another study, concentrations between 0.14 and 0.50 µg/L of PP were also found in amniotic fluid samples [84]. Furthermore, EP concentrations between 0.13 and 0.16 µg/L were found in umbilical cord blood samples, and PP between 0.21 and 0.43 µg/L and BuP between 0.04 and 0.05 µg/L were found in men and women [85]. This suggests that parabens can cross the blood–placental barrier and affect fetal development during pregnancy. In addition, there is much evidence that exposure to parabens can interfere with the homeostasis of the thyroid gland, affecting the levels of synthesis and secretion of THs in different experimental models [85,86,87].

In the following sections, we will address the main effects of parabens on the proper functioning of THs based on articles published between 1951 and 2023. The searches were carried out on the PubMed platform using the terms “thyroid” and “paraben”, and the first article directly related to the parabens was published in 1881. Table 2, Table 3 and Table 4 are organized according to the different types of parabens and their effects on the HPT axis.

### 3.1. Parabens and TSH

The pituitary gland is an endocrine gland responsible for commanding and regulating various functions in the body. Located in the midline region of the brain within the *Sella turcica*, it consists of two portions of distinct embryological origin: the adenohypophysis and the neurohypophysis. The adenohypophysis comprises the thyrotrophs, the cells responsible for the synthesis and secretion of the TSH, which act in the thyroid gland [88].

Organisms exposed to EDCs appear to have the pituitary gland as a potential target of these compounds, which can lead to growth-related disturbances, metabolic dysfunctions, and alterations in reproduction and homeostasis. Although the action of EDCs on the HPT axis is not entirely clear, it is already known that these compounds can interfere with and change the HPT axis functions [102,103].

#### 3.1.1. Parabens and TSH in Human

Hu et al. (2023) conducted a study in Wuhan, China, and found a positive association between exposure to MP in early pregnancy and a significant increase in TSH concentration in female twins [100]. An important function of TSH is to regulate the input of iodine into thyroid follicular cells. Coiffier et al. (2023) evaluated mother–child pairs from the French cohort and found an increase in TSH levels in boys and girls exposed to BuP [97]. Berger et al. (2018) demonstrated that the serum of pregnant women living in agricultural regions in Northern California presented reductions in TSH levels in association with concentrations of MP and PP in the urine samples [95]. Aker et al. (2019) found, in pregnancy samples originating from Puerto Rico, a general decrease in TSH in association with parabens detection, particularly MP, between 16–20 weeks of gestation [90]. Baker et al. (2020) studied a prospective cohort in Canada with meconium samples and found a positive correlation with the presence of MP in newborns who were later diagnosed with attention-deficit hyperactivity disorder (ADHD) [81]. The study showed that MP exposure can lead to a decrease in gestational age, a significant change in newborn weight, and a decrease in maternal TSH levels that can lead to thyroid complications in children [81]. The homeostasis and correct functioning of TSH and its target gland are important factors for the development and growth of a healthy organism, and it seems that exposure to parabens can lead to a decrease in TSH levels. There are still not many studies that describe changes in TSH levels due to exposure to parabens and a possible relationship with neurological diseases. Despite this, there are positive correlations that endocrine disruptors, such as bisphenol A and chlorpyrifos, can lead to the development of behavioral diseases and that dysregulation in the HPT axis can be a target of the agents and explain the relationships described [90,104,105]. Age, duration, and the method of exposure may potentially have an impact on how parabens can affect TSH levels. Moreover, there might be several unreported mechanisms of action that also play a role in influencing hormonal levels. Despite the poor description of the effects of TSH levels when parabens are present in humans, studies seem to indicate that parabens can change TSH levels and that it could lead to problems in human health (Figure 1).

#### 3.1.2. Parabens and TSH in Rodents

In murine models, some studies have shown how this exposure can affect the HPT axis. Gogoi, P. and Kalita, J.C. (2020) demonstrated that healthy adult female rats exposed to BuP for 7 and 21 days at low doses (1, 5, and 10 mg/kg BW/day) presented an increase in TSH levels, which caused an increase in the activity and gene expression of thyroid peroxidase and a decrease in the activity of type 1 iodothyronine deiodinase, both enzymes related to thyroid hormone biosynthesis [28]. Another study that used male rats treated for 28 days with a mixture of isopropylparaben (IPP) and isobutylparaben (IBP) (dermal exposure) at a dose of 600 mg/kg/day was able to induce a significant decrease in TSH levels in exposed animals compared to non-exposed animals. This finding shows that these compounds may have a synergistic action [94]. The dose used in this study was approximately three times greater than the known estimated level of human exposure to a single isolated paraben since it is difficult to estimate the total dose of parabens to which the human body is exposed daily [94]. Studies involving TSH and parabens are scarce, although there is some evidence that exposure to these agents leads to changes in TSH levels (Figure 2). The mechanisms involved in this pathway are not clear, and further studies are needed for a better understanding of the effects related to these agents, both in isolation and in combination, on the pituitary gland and on TSH levels. Little is discussed about the effects of exposure on the hypothalamus, and there are not many studies on how exposure to parabens affects TRH levels; perhaps exposure to parabens can affect the production and binding of TRH to its receptors as well as also affect feedback from thyroid hormones on the axis.

### 3.2. Parabens and Thyroid Hormones

THs are essential for development, growth, and metabolism, playing a key role in mammalian neurodevelopment. Therefore, alterations in the synthesis and secretion of these hormones can cause important disturbances in organisms. The literature demonstrates that exposure to parabens causes DNA damage [106], affects cell proliferative potency [106], and promotes tumorigenic processes [98], leading to a series of harmful effects on the health of individuals. Experimental studies have shown that exposure to parabens resulted in endocrine disturbances and adverse health effects. These compounds potentially bind to hormone receptors, interfering with the levels of synthesis and secretion of these hormones, relating to the increase or decrease in hormone action [107].

#### 3.2.1. Parabens and Thyroid Hormones in Human

A prospective study conducted with pregnant women between 16 and 28 weeks of gestation in Puerto Rico (PROTECT) evaluated associations between parabens and reproductive hormones and urinary paraben concentrations for quantitative analysis. Serum samples were collected at three different time points of pregnancy for measurement of sex hormone-binding globulin (SHBG), thyroid-stimulating hormone (TSH), free thyroxine (fT3), free thyroxine (fT4), and progesterone/estradiol ratio. It was shown that progesterone and estradiol increased, and SHBG showed a tendency to increase throughout pregnancy, while fT4 and fT3 decreased without changes in TSH levels, raising some concerns regarding the correct fetal development, as maternal thyroid hormones are essential for the fetus throughout pregnancy [98]. Furthermore, another study conducted in Boston demonstrated that MP was associated with an increase in fT3 and fT4 levels at 15 weeks of gestation and BuP was associated with a decrease in fT3. However, after 20 gestational weeks, MP was associated with an increase in fT3 and a decrease in fT4, indicating that the impacts of exposure to parabens on the thyroid feedback and signaling system vary according to the moment of exposure throughout pregnancy [88].

In addition, epidemiological studies have shown the relationship between thyroid hormone disbalance and parabens exposure. According to data obtained from the 2007–2008 National Health and Nutrition Examination Survey (NHANES), increased levels of EP and PP in human urine samples were associated with reductions in tT4 levels in female and male serum samples, as well as fT4 in female serum. It was also shown that the serum level of fT3 was negatively associated with EP, PP, and BuP levels in adult females but not in serum samples from males [89]. In another study, high concentrations of parabens were found in urine samples from men compared to the levels found in women’s urine samples, demonstrating that parabens can be found routinely in both men and women [103], suggesting that further association studies of various chemicals should be performed with potential common sources of exposure.

Several epidemiological studies have associated exposure to a variety of parabens found in human biological samples (ranging in concentrations between 0.1 and 38 μg/L), with disruption of thyroid function in humans, such as disruption of THs and TSH homeostasis in serum [88,89,97,98,99,108,109]. Furthermore, it has also been demonstrated that altered THs levels may be associated with an increased incidence of multiple tumors [110,111]. These data demonstrate that the effects of parabens on the THs are controversial, making it necessary to conduct more studies (experimental and epidemiological) to investigate the cellular and molecular mechanisms involved in exposure to parabens since these effects are not yet fully elucidated. In the same manner, good epidemiological studies have been conducted to establish a relationship between parabens exposure and thyroid dysfunctions, especially in pregnant women (Figure 1B).

#### 3.2.2. Parabens and Thyroid Hormones in Rodents

A study carried out with male Wistar rats evaluated the effects of exposure to BuP (50 mg/kg/day) for 60 days on the HPT axis, demonstrating an increase in TSH levels and a decrease in fT4 levels as a response to oxidative stress resulting from conversion of parabens to glutathione hydroquinone conjugates by reaction with singlet oxygen and glutathione. Furthermore, decreased fT4 production may affect the antioxidant system and contribute to the generation of oxidative stress through catalase (peroxidase) modulation [112]. The effect of paraben on the HPT axis was reinforced by another study, which demonstrated that exposure to MP and BuP (1000 mg/kg/day) for 20 days decreased T4 levels and increased thyroid mass [93]. In contrast, another study carried out with pregnant rats exposed to EP (400 mg/kg/day) and BuP (200 and 400 mg/kg/day) for 14 days did not demonstrate significant changes in maternal or newborn THs serum levels [86].

#### 3.2.3. Parabens and Thyroid Hormones in Vertebrates

A study with Zebrafish larvae showed that exposure to EP (0, 20, 50, and 100 μM), PP (0, 5, 10, and 20 μM), BuP (0, 2, 5, and 10 μM), and MP (0, 20, 100, and 200 μM) between 2 and 120 h post-fertilization (hpf) was able to decrease THs concentrations in most tested concentrations, and this exposure also showed a negative correlation with the survival rate of larvae. Furthermore, it has also been demonstrated that the toxicity of parabens increased according to the length of the alkyl carbon chain group, and the order of toxicity was BuP > PP > EP > MP (Figure 3A). These results reinforce the effects of exposure to these compounds on organisms in general, leading to a series of deleterious effects even on zebrafish larvae and affecting correct development by interfering with the correct synthesis and secretion of THs, as THs are fundamental for the processes of proliferation, migration, differentiation, and neuronal signaling, as well as brain myelination during neurodevelopment. Any interference in the levels of THs at this stage of development would have serious consequences for neurodevelopment, leading to several morphofunctional and physiological consequences, including the possibility of affecting the juvenile development process of these animals [91].

In another experimental model, the authors evaluated the effects of exposure to environmental pollutants on tadpole metamorphosis. In this study, no significant differences were observed in the metamorphosis of tadpoles (between 12- and 14 days post fertilization) exposed to low concentrations of PP (5 mg/L) for 14 days and control animals. However, the authors observed a high mortality rate of tadpoles exposed to the highest concentration of PP (12.5 mg/L), revealing an acute toxic effect at increased concentrations of PP. In addition, a decrease in PP concentrations was also observed in the water after two days of exposure, indicating its rapid absorption (Figure 3B). This suggests that prolonged exposure to this compound may result in changes in the endocrine system of these individuals, even at low concentrations [87].

## 4. Conclusions

The studies presented in this review provide evidence, both from epidemiological and experimental models, of an association between paraben exposure and HPT axis dysfunctions. Previous studies have already correlated exposure to EDCs and disturbances in this axis. Although parabens are a class of EDCs widely used in industry, there are few studies that describe their effects on the HPT axis. We first pooled studies describing pituitary effects resulting from exposure to parabens. Although scarce, these studies described distinct effects. The data on humans are conflicting; however, there is a greater number of studies describing a decrease in TSH levels in humans compared to those that found increased TSH levels. In rodents, exposure appears to decrease TSH levels. Perhaps age, duration, dose, and mode of exposure may influence how parabens can affect TSH levels. Although the HPT axis functions similarly in rodents and humans, the sensitivity to paraben exposure can vary. Therefore, more investigation should be done to understand this discrepancy. We must also consider the possibility that the dose, timing, tissue response, and other variables may have altered these species-specific reactions.

Additionally, there could be various other yet undescribed mechanisms of action that influence the effects on hormone levels. Effects on THs levels are also poorly described. However, recent studies have shown different actions between different types of parabens on the HPT axis. While EP, PP, and BuP were associated with THs decrease, MP was associated with THs increase. This only allows us to speculate that the mechanisms of action of these parabens are different. Furthermore, studies of exposure to parabens are more evident in women and scarce in men. Women are more likely to have thyroid disease than men, and this difference may be due to the female thyroid gland having a higher amount of oxidative stress [59,113]. Additionally, a variety of thyroid-related processes, including iodide uptake, thyroperoxidase activity, and hydrogen peroxide production, which are necessary for the synthesis of thyroid hormones, seem to be impacted by gender and estrogen [114]. Compared to the male thyroid, the female thyroid is more susceptible to estrogen. In contrast to their male counterparts, adult female rats produce more reactive oxygen species (ROS) under the control of estrogen. Finally, disruption of estrogenic and androgenic receptors was caused by methylparaben, indicating that parabens may have estrogenic effects [93,115,116].

Therefore, future studies are needed to clarify the effects of exposure to different parabens and their mechanisms of action on the HPT axis since, between 1951 and 2023, few studies were published that show the effects of exposure to these compounds, and the mechanisms of action, as well as the physiological and/or molecular effects of these compounds in the organism, are still not fully known. To establish/characterize the effects of parabens exposure on human health in more detail, we will require more epidemiological research that makes use of large human cohorts from various world areas and life stages. Finally, to prevent thyroid illnesses, we recommend that regulatory agencies and WHO work more effectively to limit human exposure to parabens.

## Figures and Tables

**Figure 1 ijms-24-15246-f001:**
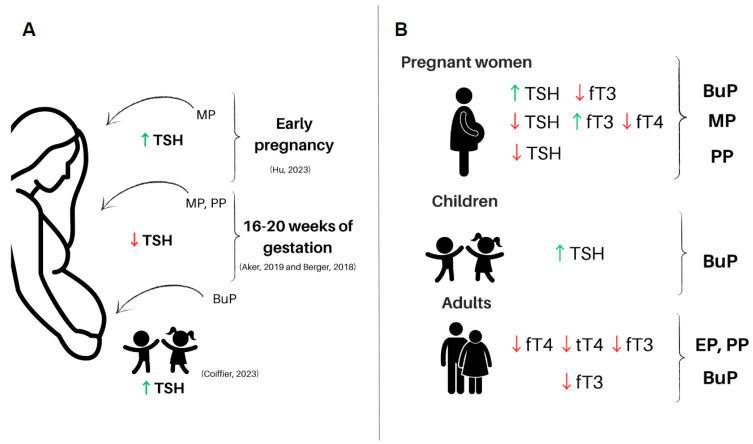
Main effects of exposure to parabens on human health [90,95,97,100]. (**A**) Effects of paraben exposureduring pregnancy on the HPT axis hormones and maternal and newborn health. (**B**) Effects of general human exposure to parabens on the HPT axis hormones. Legend: BuP—butylparaben; EP—ethylparaben; MP—methylparaben; PP—propylparaben; TSH—thyroid-stimulating hormone; THs—thyroid hormones; fT4—free thyroxine levels; fT3—free triiodothyronine levels; tT4—total thyroxine levels; tT3—total triiodothyronine levels. This figure was made using the Canva platform.

**Figure 2 ijms-24-15246-f002:**
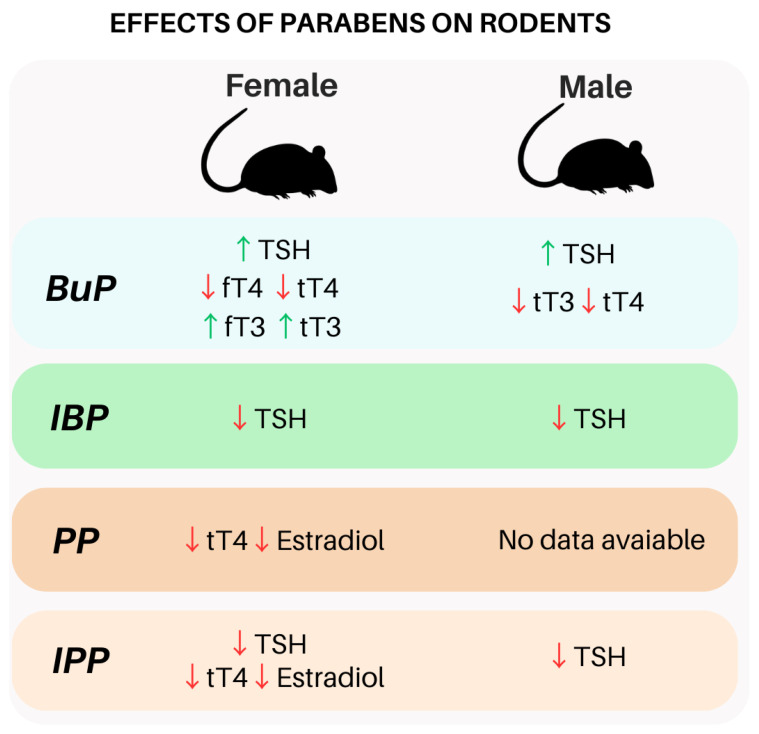
Main effects of exposure to parabens on rodents. Effects of parabens exposure in the HPT axis of female and male rodents. Legend: BuP—butylparaben; IBP—isobutylparaben; IPP—isopropylparaben; PP—propylparaben; TSH—thyroid-stimulating hormone; fT4—free thyroxine levels; fT3—free triiodothyronine levels; tT4—total thyroxine levels; tT3—total triiodothyronine levels. This figure was made using the Canva platform.

**Figure 3 ijms-24-15246-f003:**
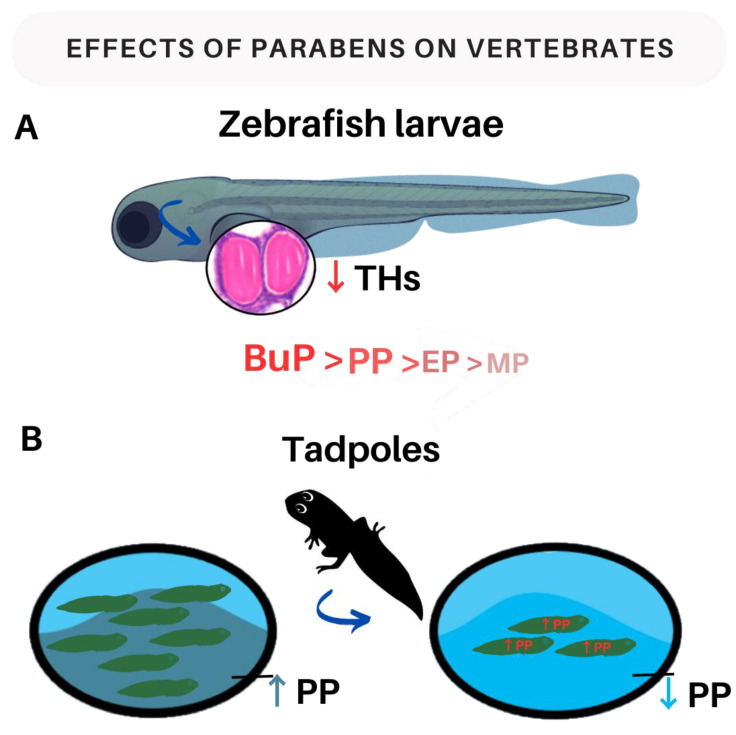
Main effects of exposure to parabens on vertebrates. (**A**) Consequences of paraben exposure in zebrafish larvae. The blue arrow indicates the effects of paraben exposure on the larvae’s thyroid gland, resulting in a decrease in thyroid hormones (THs). The studied parabens are listed below the larva in order of toxicity: BuP > PP > EP > MP. (**B**) Impacts of paraben exposure on tadpoles. On the left side of the image, we can see water with high levels of propylparaben (PP). Tadpoles exposed to parabens in the water experience a significant mortality rate, as shown on the right side of the image. The decrease in PP levels in the water indicates that tadpoles absorb PP from the water, accumulating the substance in their bodies. This figure was made using the Canva platform. Legend: THs—thyroid hormones; BuP—butylparaben; PP—propylparaben; EP—ethylparaben; MP—methylparaben.

**Table 1 ijms-24-15246-t001:** Chemical characteristics of parabens.

Class	No. CAS	Molecular Weight (g/mol)	Chemical Formula	Chemical Structure
Benzylparaben	94-18-8	228.2433	C_14_H_12_O_3_	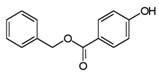
Butylparaben	94-26-8	194.2271	C_11_H_14_O_3_	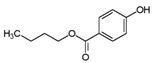
Ethylparaben	120-47-8	166.1739	C_9_H_10_O_3_	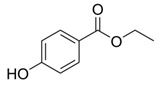
Methylparaben	99-76-3	152.1473	C_8_H_8_O_3_	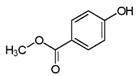
Propylparaben	94-13-3	180.2005	C_10_H_12_O_3_	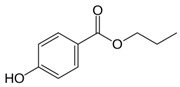

**Table 2 ijms-24-15246-t002:** Effects of butylparaben on thyroid function.

Model	Exposure/Dose/Analyses	Main Results	References
Human (men)	Serum hormone analysis of Inhibin, FSH, LH, testosterone, estradiol, TSH, T3, and T4.	BuP was associated with ↑ TSH, T4, fT4 after 96 h of exposure.	[66]
Pregnant women (12–14 weeks)	Urine collection at 3 different gestational moments (16–20, 20–24, 24–28 gestation weeks) and hormone analyses.	BuP was associated with ↑ estradiol and progesterone with ↓ fT3 and fT4 at visit 3. tT3 and TSH levels did not change between visits.	[88]
Pregnant women—Boston (>15 weeks)	Collection of urine and blood at 4 different gestational moments (9, 17, 26, and 35 weeks of gestation).	BuP was associated with ↓ T3, ↓ T3/T4 ratio, and ↑ TSH.	[89]
Pregnant women—Puerto Rico	Urine collection at 3 different gestational moments (16–20, 20–24, 24–28 weeks of gestation).	Exposure to BuP has been associated with ↓ SHBG.	[90]
Pregnant women	Collection of maternal urine on the day of delivery and collection of umbilical cord blood for hormone measurement.	BuP is associated with ↑ boys’ body weight at birth.	[72]
Male Wistar rats	Oral exposure BuP (10 mg/kg/day), BuP (50 mg/kg/day), and BuP+TCS (triclosan) (50 mg+10 mg/kg/day) for 60 days.	BuP (50 mg/kg/d) was associated with ↑ TSH and ↓ T3 and T4.	[91]
Female Wistar rats	Subcutaneous administration of BuP at doses of 1, 5, and 10 mg/kg/day for 7 and 21 days.	↑ TSH in BuP1 at 7 and 21 days; ↓ fT4 and tT4 at all concentrations (7 and 21 days); ↑ fT3, tT3, and TPO in SP1 and SP5 at 7 and 21 days.	[28]
Zebrafish larvae	Larvae were exposed to the following concentrations: 0, 2, 5, and 10 μM of BuP.	↓ T4 levels at most concentrations tested (BuP 5 and 10 μM) and ↓ T3 levels at all concentrations tested. That exposure also led to an increase in TSH gene expression at all concentrations of BuP.	[92]

**Table 3 ijms-24-15246-t003:** Effects of benzylparaben, isopropylparaben, isobutylparaben, and propylparaben on thyroid function.

Model	Exposure/Dose/Analyses	Main Results	References
Female Sprague Dawley rats	Oral exposure to MP, EP, PP, isopropylparaben (IPP), BuP, and isobutylparaben (IBP) (62.5; 250 and 1000 mg/kg/day) from the 21st to the 40th postnatal day.	PP and IPP were associated with ↓ T4 and estradiol and changes in thyroid weight.	[93]
Male and female Sprague Dawley rats	Injections of IPP, IBP, or mixture of IPP and IBP at 50, 100, 300, and 600 mg/kg bw dissolved in 100 mL of ethanol (99%), 5 days per week for 28 days.	The mixture of IPP and IBP induces a decrease of TSH in exposed individuals at an exposure of 600 mg/kg bw.	[94]
Pregnant women (PROTECT)	Blood collection at two different gestational moments for measurement of SHBG, TSH, fT3, fT4, and progesterone/estradiol ratio; urine collection for detection of phenols and parabens by high-performance liquid chromatography (HPLC).	↑ estradiol and progesterone at the last visit; ↓ fT3 and fT4 at the last visit with no changes in TSH levels.	[88]
Pregnant women—Puerto Rico	Urine and blood collection at 4 time points during pregnancy. Parabens were detected in urine by chromatography. In the blood, tT4, fT4, TSH, and T3 were measured.	PP was inversely associated with fT4.	[89]
Pregnant women—California	Urine and blood collection in the second gestational trimester and blood collection from neonates for measurement of tT4 and TSH.	PP was inversely associated with TSH levels with no changes in tT4 levels.	[95]
Pregnant women—Puerto Rico	Urine collection at 3 different gestational moments (16–20, 20–24, 24–28 weeks of gestation).	Exposure to PP was associated with ↓ SHBG and T3/T4 ratio.	[90]
Human (population of Wuhan, China)	Urine collection and detection of MP, EP, and PP.	PP has been associated with an increased risk of thyroid cancer.	[96]
Newborn human	Newborn blood spots were collected as part of the neonatal screening program, TSH and tT4 were assessed using immunofluorescence.	BuP increased TSH and decreased T4 hormone levels have been demonstrated in newborns and women with less than 150 μg/L of iodine.	[97]
Amphibian tadpoles	Oral exposure to PP (0.05; 0.5 and 5 mg/L) for 14 days.	An increase in PP concentrations in water has been associated with an acute toxic effect.	[87]
Zebrafish larvae	Larvae were exposed to the following concentrations: 0, 5, 10, and 20 μM of PP.	Serum T3 and T4 concentrations decreased at all concentrations tested. In 10 and 20 μM groups, PP increases TSH gene expression.	[92]

**Table 4 ijms-24-15246-t004:** Effects of ethylparaben and methylparaben on thyroid function.

Model	Exposure/Dose/Analyses	Main Results	References
Human	Urine samples were collected from patients at Wuhan Central Hospital who had thyroid disease and required surgery. Some types of parabens were detected in these samples, such as MP, EP, and PP.	MP and EP were found in urine samples in 99.06%, 95.29%, and 92%, respectively. There was a ↑ concentration of all parabens in the urine of both the nodule and cancer groups. MP and EP were associated with a benign nodule, especially when in higher concentrations. All three parabens studied were associated with an increased risk of thyroid cancer, with EP having the greatest association.	[96]
Mother—children	Urine samples from mothers of newborns were collected on the day of delivery. The concentrations of 5 parabens were determined by chromatography. Umbilical cord blood was collected immediately after birth, in which tT3, tT4, fT3, fT4, TSH, anti-TPO, and anti-TG were measured.	MP and EP were detected in the urine of the evaluated mothers. EP was positively related to increased tT3 in the umbilical cord and to anti-TPO. EP was correlated with increased birth weight in boys, but not in girls.	[72]
Human—Korea	Population study with 1254 people from Korea. Urine samples from this population were collected for analysis of the presence of EDC. Blood serum samples were also collected for measurement of tT4 and fT4, tT3 and fT3, TSH, anti-TPO, anti-thyroglobulin, thyroxine-binding globulin (TBG), and iodothyronine deiodinase (DIO) activity.	Parabens were found in most of the studied population (more than 90%). MP showed a positive association with altered levels of tT3. The increase in MP and EP parabens was correlated with an increase in TBG.	[59]
Pregnant women—Puerto Rico	Urine collection at 3 different gestational moments (16–20, 20–24, 24–28 weeks of gestation).	MP was associated with a decrease in SHBG. MP leads to a significant decrease in TSH and a decrease in the T3/T4 ratio particularly at weeks 24–28 of gestation.	[90]
Pregnant women—California	Urine and blood collection in the second gestational trimester and blood collection from neonates for measurement of tT4 and TSH.	MP was inversely associated with TSH levels with no changes in tT4 levels.	[95]
Pregnant women—Puerto Rico	Urine and blood collection at 4 time points during pregnancy. Parabens were detected in urine by chromatography. In the blood, tT4, fT4, TSH, and T3 were measured.	Urine samples that tested positive for the presence of MP were associated with increased T3 and negatively associated with fT4 at gestational age less than 21 weeks.	[89]
Pregnant women (12–14 weeks)	Urine collection at 3 different gestational moments (16–20, 20–24, 24–28 gestation weeks) and hormone analyses.	MP (293 ng/mL) was associated with a 7.70% increase in SHBG.	[98]
Human	Urine samples from a representative portion of the US population to assess urinary concentrations of triclosan and parabens.	Inverse associations have been found between parabens and circulating levels of thyroid hormones in adults, where women appear to be more vulnerable to exposure.	[99]
Mother–children	Maternal blood was collected during the first prenatal care visit for TSH measurement. MP was detected in meconium samples from newborns.	MP exposure leads to a decrease in gestational age, a significant change in newborn weight, and a decrease in maternal TSH levels. In addition, MP in meconium was associated with about a 16% decrease in tT3 and a decrease in fT4. MP may influence maternal thyroid physiology during pregnancy, and this may lead to the development of ADHD.	[81]
Mother–twin pairs	MP was extracted from urine samples of pregnant women using liquid-liquid extraction. Neonatal TSH levels were abstracted from medical records in China.	MP exposure in early pregnancy was associated with an increased intra-twin TSH difference.	[100]
Wistar rats	Oral exposure for 90 days to BPA (50 mg/kg) or BPA+MP (250 mg/kg).	A minimal thyroid receptor antagonistic effect was only observed after treatment with BPA+MP. MP demonstrated antioxidant properties by reducing lipid peroxidation and generation of hydroxyl radicals induced by exposure to BPA.	[101]
Zebrafish larvae	Larvae were exposed to the following concentrations: 0, 20, 50, and 100 μM of EP and 0, 20, 100, and 200 μM of MP.	Serum T3 concentrations decreased at most concentrations tested (EP at 50, 100 μM and MP at 20, 100, and 200 μM) and T4 concentrations decreased at all concentrations tested.	[92]

## Data Availability

The images in this study were generated using Canva for Education (www.canva.com).

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
