# Peer review of "Environmental Endocrinology: Parabens Hazardous Effects on Hypothalamic–Pituitary–Thyroid Axis"

_ijms, 2023, doi:10.3390/ijms242015246_

Round 1

Reviewer 1 Report

The authors in the paper considered research performed in vitro, in vivo, and epidemiological studies published between 1951 and 2023, which demonstrated an association between paragons exposure and HPT axis dysfunction. The work is very interesting and the topic of great topically. The work is well organized and comprehensively written. The work is scientifically sound and not misleading, the bibliography is appropriate.

Author Response

Comments and Suggestions for Authors

The authors in the paper considered research performed in vitro, in vivo, and epidemiological studies published between 1951 and 2023, which demonstrated an association between paragons exposure and HPT axis dysfunction. The work is very interesting and the topic of great topically. The work is well organized and comprehensively written. The work is scientifically sound and not misleading, the bibliography is appropriate.

Response: Thank you very much for your consideration.

Reviewer 2 Report

The paper is interesting but it requires major revision before publication. Please find the detailed comments in the attached file.

Kind regards,

Reviewer

English language is correct, I've found only some minor editorial errors that should be corrected before publication.

Author Response

Comments and Suggestions for Authors

The paper „Environmental Endocrinology: Parabens hazardous effects on HPT axis” is a review of current knowledge concerning the influence of a selected group of endocrine disrupting chemicals (namely parabens) on thyroid function, and therefore it may be interesting for the readers of “International Journal of Molecular Sciences”. However, in my opinion, the manuscript requires major revision before publishing. The concept, of the paper is – in general – good, unfortunately there are some issues that require correction or even re-writing.

  1. The title of the paper – I propose do not use the abbreviation that is not necessarily obvious for the readers in the title (in medicine HPT is frequently used for hypoparathyroidismus)

Response: Thank you for this suggestion. We rewrote the title of the revised manuscript.

  1. Affiliations - please ensure that affiliations may be in other language than English, as now only a part of affiliation 3 is in English

Response: Thank you for noting your mistake. We performed the correct changes. fixed that mistake.

  1. Graphical abstract – it’s valuable to provide a graphical abstract, however the presented one is definitely unclear, due to numerous unexplained abbreviations and – quite surprisingly - question mark next to the arrows indicating relationships between pituitary gland and thyroid; please improve this picture to make it possible to understand before writing the paper.

Response: Thank you for this suggestion. We added a legend to the Graphical Abstract and fixed the figure.

  1. Key words should be reconsidered – please do not use unexplained abbreviation TH (that in pediatric endocrinology is routinely used for “target height”) or HPT (hypoparathyroidismus?); key word “hormone” is definitely too general

Response:. We rewrote all keywords. Thank you.

  1. Abbreviations – please unify the abbreviation used in the whole paper and introduce all abbreviation when used for the first time, e.g.
    • HPT is explained in abstract as “hypothalamic-pituitary-thyroid” (line 44), while in the main text as “hypothalamus-pituitary-thyroid” (line 65) – please unify
    • free thyroxine is abbreviated as either fT4 or FT4, total thyroxine as tT4 or TT4
    • free triiodothyronine is abbreviated as either fT3 or FT3, total triiodothyronine as tT3 or TT3
    • using two abbreviations EDC and EDCs for endocrine-disrupting chemicals seems not to be justified (as the authors do not use it in the context of one selected disruptor); the same relates to abbreviation “THs” (line 356)
    • “thyroid hormones” (TH) is introduced for the first time in line 71, and for the second time – unnecessarily – in line 128
    • “thyroglobulin (TG) should be introduced in line 124, not in line 129
    • please explain the abbreviation DUOX (line 141)
    • please use the abbreviations that are introduced (e.g. PP for parabens in lines 196 and 202)
    • please explain the abbreviation “EFSA” (line 200)
    • abbreviation IPP for isopropylparaben in Table 3 should be explained while used for the first time, while citing Vo, 2010, not at the second paper of Kim, 2015
    • please explain the abbreviation HPLC (Table 3, last line in the content related to the paper of Wang, 2015)
    • please explain the abbreviation PE in Table 3 in last line of description of the paper of Wu, 2022
  • “thyroid binding globulin” should have abbreviation TBG (not TGB) – please correct (e.g. in Table 4, reference to paper of Choi, 2020 at page 11; the abbreviation should be explained at first use
  • please explain the abbreviation “SHGB” in Table 4, reference to Aker, 2019, if it relates to sex hormone binding globulin, it should be “SHBG”
  • please explain the abbreviation DIO in Table 4 in reference to paper of Choi, 2020 at page 11
  • please explain abbreviation BP and use previously introduced abbreviation BeP in Table 3 in last description on page 10, related to the paper of Li, 2020 (abbreviations BeP and Bup have been introduced in line 177)
  • do not introduce the same abbreviations few times and consequently use the introduced abbreviations where appropriate

Response: Thank you for this suggestion. All abbreviations were reviewed in the whole revised manuscript.

  1. Chapter numbering – it seems that chapter numbering requires modification
    • sections 2.1 to 2.8 seem not to be the subsection of section 2. “Thyroid morphophysiology: an overview”; it would be better to re-number section “Parabens” as 3.0 and 2.2 to 2.8 as its subsections
    • It seems that last sentence of subsections 2.3 “Parabens and TSH in rodents” (line 269) is in fact a title of subsection 2.4 and should be placed after a number of subsection and written in italics; consequently, a paragraph starting from 2.4 (lines 270-288) is not a title of the chapter and should not be in italic
    • the most appropriate could be the following numbering:
  2. Introduction
  3. Thyroid morphophysiology: an overview
  4. Parabens
    • Parabens and TSH
      • Parabens and TSH in human
      • Parabens and TSH in rodents
    • Parabens and Thyroid Hormones
      • Parabens and Thyroid Hormones in human
      • Parabens and Thyroid Hormones in rodents

3.2.2. Parabens and Thyroid Hormones in vertebrates

  1. Conclusion (or better “Conclusions”)

Response: Thank you for these suggestions. We corrected the numbering.

  1. Substantive remarks
    • please comment the fact that – despite searching in PubMed for publications since 1957 -the earliest cited publication on parabens is from 1997 (few earlier papers that are cited seem not to be related directly to the topic of the paper)

Response: Although we searched for studies between the years 1951 and 2023 related to parabens and thyroid function, the first article found directly related to parabens reported throughout the review was only published after 1951.

  • blood vessels in thyroid gland are not “responsible for the distribution of hormones to the body” (line 118), please re-write

  • it’s known that a precursor of thyroid hormones is tyrosine, please add appropriate citation that clearly confirms the statement concerning TG: “This precursor protein of TH …” (line 132)
  • in Table 2, the citation of paper of Aker, 2016 provides no information concerning the effects of buthylparaben, please re-write or remove; the same refers to Table 3, the citation of paper of Wang, 2015

Response: In Table 2, the paper of Aker 2016 provides information concerning exposure to various parabens and other compounds (that are not the focus of this study). However, among the parabens tested, only butylparaben showed significant differences in the observed results, so this paper was added to this table. In Table 3, there was an error in the citation. The correct paper would be Aker's, 2016. We have corrected it in the table. Thank you for the suggestions.

  • please provide more detailed information in Table 4 in comment to paper of Aker, 2016 (page 11), what dose or concentration of MP was associated with exactly 7.7% increase of SHBG?

Response: The author does not provide the information about dose or concentration in the cohort studied.

  • in Table 4 in a description related to the paper of Genuis, 2013 (1st at page 12), there is no information concerning the effects of parabens on thyroid gland – please re-write or remove this citation from the Table
  • in section “Parabens and TSH in human” there is conflicting information that “… exposure to parabens can lead to a decrease in TSH levels” (lines 260-261) and “… parabens can increase TSH levels” (lines 267-268), with no appropriate references – please verify these statements, provide appropriate references for them and try to comment this situation; moreover, in Conclusion section there is only information that in human’s the exposure to parabens may lead to increase in TSH levels (lines 373-375) that is not fully in line with previous descriptions

Response: Thank you for the suggestions. We corrected these topics.

 Tables – there are some necessary corrections

    • it seems that the Tables have not been prepared with use of dedicated Template, as fonts are too large and different from these used in the Journal, please use exactly the template provided by IJMS in “Instructions for Authors”
    • please ensure that the titles of Tables are placed over particular Tables, not at the bottom of previous page (e.g. a title of Table 1 is at the bottom of page 5 in line 188, while the whole Table is on page 6)
    • Table 2 should start on page 7 (now this page is almost empty)
    • please provide number of references in Tables 2 ,3 and 4 as now it is difficult to identify appropriate references
    • titles of Tables 3 and 4 should start from “Effects of …” like for Table 2
    • Table 3, citation of Berger, 2018 – the sentence starting from “Exposure to propylparaben … “ seems explain methodology, not results, please also use the abbreviation PP
    • it could be helpful to provide a list of abbreviations used in the whole paper

Response: We have corrected all these topics. Regarding abbreviations, we made sure that all of them were mentioned and described throughout the text. Thank you for the suggestions.

  1. Grammar and editorial corrections – please re-write the following sentences:
    • “TRH acts in their receptor in the pituitary” (line 69)
    • … “and the development of TD focus on human … (line 112)
    • “After absorption, parabens are absorbed from gastrointestinal tract …” (line 183)
    • “Puerto in Rico” (line 252)
    • please remove unnecessary initials of authors in line 270
    • line 369/370 – it would be better to write about “HPT axis dysfunctions
    • line 386 – it would be better to write about “thyroid diseases
    • there are also few minor editorial errors that should be corrected

Response: Thank you for the suggestions. We corrected all these topics.

  1. Measurement units and missing data please confirm the units for PP concentrations: mg/L in line 13, while µg/L in line 217 (the second one are a thousand times lower than the first); please place appropriate references directly after cited values, as it is impossible to identify the source of particular data when six references is provided together for the whole paragraph (in line 217)

  • please ensure if values provided in Table 2 in the last citation (Lang, 2022) are concentrations, as they seem to be expressed like doses (in µM while not in µM/L); the same refers to the last citation in Table 3 (Coiffier, 2023)

  • line 317, please explain which levels of EP and PP were associated with reductions in TT4 (use appropriate abbreviation)
  • lines 328-329, shouldn’t be “homeostasis in serum”?

Response: Thank you for the suggestions. We corrected all these topics.

  1. References should be prepared according to the Instructions for Authors, provided by MDPI for the Journal, please prepare them accordingly; provide the year of publication for Ref. 56.

Response: Thank you for the observation. We corrected the references in the tables.

Reviewer 3 Report

Review on the manuscript of Azeredo, D. B. C. et al.: “Environmental Endocrinology: Parabens hazardous effects on HPT axis”.

In this manuscript, authors present a review on hazardous effects of parabens on Hypothalamus-Pituitary-Thyroid axis, in humans and other animal models.

The manuscript is very clear and well written. In addition, I consider that the manuscript is precise on the question that authors proposed to review. Thus, the issues that arise to me are listed below, so, I hope authors find the following comments and suggestions useful.

1 – In humans, TSH levels are increased in response to parabens’ exposure, whereas, in rodents, parabens’ exposure decreases TSH levels. Do authors have explanations for such effect?

2 – Do authors have any hypothesis why studies on parabens are more evident in woman than in men?

3 – The topic 2.1 “Parabens, in this version of the manuscript is a subtopic of the topic 2 “Thyroid Morphophysiology: an overview”. Do authors consider to make the topic Parabens as a topic 3 (it has nothing related to the topic 2)?

4 – Authors show the actual scenario of regulation applied to parabens in Brazil and European Union. Can authors provide information on the regulation of parabens in the USA?

5 – In table 2, can authors show the results from human studies first and then the studies with rodents (the data from the Taha, 2020 study is in the middle of 2 human studies)?

6 – The legend of table 3 indicates that the table summarizes studies on benzylparaben and propylparaben on thyroid function. However, in the content, data from methylparaben, ethylparaben, isopropylparaben, butylparaben and isobutylparaben are also provided. Can authors correct the legend of that table? The same is valid for table 4, where data from other parabens than ethylparaben and methylparaben are provided.

7 – Can authors correct the format of topic 2.4 “Parabens and TSH in rodents” to the same text style?

8 – I recommend authors to include in the conclusion section some future perspectives on this topic. For instance, what should be done to characterize in more detail the consequences of parabens’ exposure to human health.

9 – I encourage authors to make some figures to be incorporated into the manuscript (3-4 figures would be good). Without figures, the manuscript becomes less attractive to the readers. I also encourage authors to improve the quality of the graphical abstract (for instance, indicate the adenohypophysis and say that this part of the pituitary gland is responsible for TSH synthesis and release).

Author Response

Review on the manuscript of Azeredo, D. B. C. et al.: “Environmental Endocrinology: Parabens hazardous effects on HPT axis”.

In this manuscript, authors present a review on hazardous effects of parabens on Hypothalamus-Pituitary-Thyroid axis, in humans and other animal models.

The manuscript is very clear and well written. In addition, I consider that the manuscript is precise on the question that authors proposed to review. Thus, the issues that arise to me are listed below, so, I hope authors find the following comments and suggestions useful.

1 – In humans, TSH levels are increased in response to parabens’ exposure, whereas, in rodents, parabens’ exposure decreases TSH levels. Do authors have explanations for such effect?

Response: The HPT axis in humans and rodents has similarly, however the sensitivity of the axis to paraben exposure must vary. Therefore, more investigation should be done to understand this discrepancy. We must also take into account the possibility that the dose, timing, and other variables may have altered these species-specific reactions.

This sentence was included in the “Conclusion” section.

2 – Do authors have any hypothesis why studies on parabens are more evident in women than in men?

Response: Women are more likely to have thyroid disease than men, and this difference may be due to the female thyroid gland having a higher amount of oxidative stress. Additionally, a variety of thyroid-related processes, including iodide uptake, thyroperoxidase activity, and hydrogen peroxide production, which are necessary for the synthesis of thyroid hormones, seem to be impacted by gender and estrogen. Compared to the male thyroid, the female thyroid is more susceptible to estrogen. In contrast to their male counterparts, adult female rats produce more reactive oxygen species (ROS) under the control of estrogen. Finally, disruption of estrogenic and androgenic receptors was caused by methylparaben, indicating that parabens may have estrogenic effects .

This sentence was included in the “Conclusion” section.

References:

1- Terasaka et al. (2006). Expression profiling of estrogen-responsive genes in breast cancer cells treated with alkylphenols, chlorinated phenols, parabens, or bis- and benzoylphenols for evaluation of estrogenic activity. Toxicology Letters. doi:https://doi.org/10.1016/j.toxlet.2005.10.005.

2- Vo et al. (2010). Potential estrogenic effect(s) of parabens at the prepubertal stage of a postnatal female rat model. Reprod Toxicol. doi: 10.1016/j.reprotox.2010.01.013.

3- Fortunato et al., 2012. Sexual Dimorphism of Thyroid Reactive Oxygen Species Production Due to Higher NADPH Oxidase 4 Expression in Female Thyroid Glands. Thyroid. 10.1089/thy.2012.0142.

4-. Costa et al., 2017. Endocrine-disrupting effects of methylparaben on the adult gerbil prostate.Environ Toxicol. doi: 10.1002/tox.22403.

5- Miranda el al. (2020). Thyroid redox imbalance in adult Wistar rats that were exposed to nicotine during breastfeeding. Scientific reports. 10.1038/s41598-020-72725-w.

5-. Nowak et al. (2021). Methylparaben-induced regulation of estrogenic signaling in human neutrophils. Molec and Cell Endocrinology.doi:10.1016/j.mce.2021.111470

3 – The topic 2.1 “Parabens, in this version of the manuscript is a subtopic of the topic 2 “Thyroid Morphophysiology: an overview”. Do authors consider to make the topic Parabens as a topic 3 (it has nothing related to the topic 2)?

Response: Thank you for this suggestion. We corrected the numbering.

4 – Authors show the actual scenario of regulation applied to parabens in Brazil and European Union. Can authors provide information on the regulation of parabens in the USA?

Response: Preservatives used in cosmetics are not subject to any specific Food Drug Administration (FDA) regulations. Preservatives in cosmetics are treated by the law in the same way as other cosmetic additives. Under the Federal Food, Drug, and Cosmetic Act (FD&C Act), cosmetic products and ingredients, other than color additives, do not need FDA approval before they go on the market. However, if a cosmetic has been adulterated or misbranded, it is illegal to promote it in interstate commerce. For instance, this means that cosmetics must be appropriately labeled and safe for consumers to use as directed on the label or in the conventional manner. FDA has the authority to take action against a cosmetic product on the market that violates the regulations we uphold. However, in order to take action against a cosmetic for safety grounds, we must have solid scientific evidence demonstrating that the product is dangerous when used as directed by the label or in the usual manner by consumers.

Reference: https://www.fda.gov/cosmetics/cosmetic-ingredients/parabens-cosmetics#regulations

5 – In table 2, can authors show the results from human studies first and then the studies with rodents (the data from the Taha, 2020 study is in the middle of 2 human studies)?

Response: Thank you for this observation. We inserted the rodent studies after the human results.

6 – The legend of table 3 indicates that the table summarizes studies on benzylparaben and propylparaben on thyroid function. However, in the content, data from methylparaben, ethylparaben, isopropylparaben, butylparaben and isobutylparaben are also provided. Can authors correct the legend of that table? The same is valid for table 4, where data from other parabens than ethylparaben and methylparaben are provided.

Response: We corrected the text formatting. Thank you for this observation.

7 – Can authors correct the format of topic 2.4 “Parabens and TSH in rodents” to the same text style?

Response: We corrected the text formatting. Thank you for this observation.

8 – I recommend authors to include in the conclusion section some future perspectives on this topic. For instance, what should be done to characterize in more detail the consequences of parabens’ exposure to human health.

Response: Thanks for this observation.

“In order to establish/characterize the effects of parabens exposure on human health in more detail, we will require more epidemiological research that make use of large human cohorts from various world areas and life stages.”

This sentence was included in the “Conclusion” section.

9 – I encourage authors to make some figures to be incorporated into the manuscript (3-4 figures would be good). Without figures, the manuscript becomes less attractive to the readers. I also encourage authors to improve the quality of the graphical abstract (for instance, indicate the adenohypophysis and say that this part of the pituitary gland is responsible for TSH synthesis and release).

Response: We agree with you. Figures 3 and 4 were added to the manuscript with their respective descriptions.

Reviewer 4 Report

The comments are in the paper.

Author Response

Comments and Suggestions for Authors

  • Add a schematic representation of the effect of parabens on vertebrates;

Response: The schematic representation was added.

  • Describe the effects of THs synthesis and secretion on the correct development in zebrafish larvae;

Response: THs are fundamental for the processes of proliferation, migration, differentiation and neuronal signaling, as well as for brain myelination during neurodevelopment. Any interference in the levels of THs at this stage of development would have serious consequences for neurodevelopment, leading to several morphofunctional and physiological consequences, including the possibility of affecting the juvenile development process of these animals.

We insert this information into the text. Thank you for the suggestion.

  • Add symbols after conclusion

Response: All symbols were added.

Round 2

Reviewer 2 Report

Thank you for resubmission a revised version of the paper „Environmental Endocrinology” and all explanations. Nevertheless, there are still some issues that require further corrections and the next round of review is necessary in my opinion.

A.    Graphical abstract – unfortunately, according to the Instructions for Authors, Graphical abstract should be an image, so, all text should be included within this image, e.g. abbreviations should be explained in the space of this image, without additional text that is definitely more expanded than just the legend – please correct accordingly;

B.    Abbreviations

-         please unify abbreviations in Graphical abstract, Abstract and and the main text (e.g. TH, HT, THs)

-         abbreviation “HT”, used il line 9 of Abstract, is still not explained

-         abbreviations: HPT, PPI, THs are explained over the Graphical abstract but not used in Figure;  instead, abbreviations TRH, IBP and TH are used in Figure but not explained – please correct more carefully

-         please explain the abbreviation “SHGB” in Table 4, reference to Aker, 2019, if it relates to sex hormone binding globulin, it should be “SHBG” – not corrected after first review

-         please explain the abbreviation “PM” on page 7 in 5th line from the top

C.    Substantive remarks

-         I understand the answer of the Authors, however, the information that – despite searching in PubMed for publications since 1951 -the earliest cited publication on parabens is much younger – should also be placed in the main text  to indicate, when really were published the papers included

-         in Table 2, the citation of paper of Aker, 2016 provides no information concerning the effects of buthylparaben, please re-write or remove; the same refers to Table 3, the citation of paper of Wang, 2015

Response: In Table 2, the paper of Aker 2016 provides information concerning exposure to various parabens and other compounds (that are not the focus of this study). However, among the parabens tested, only butylparaben showed significant differences in the observed results, so this paper was added to this table. In Table 3, there was an error in the citation. The correct paper would be Aker's, 2016. We have corrected it in the table. Thank you for the suggestions.

New comment: if the paper provides information related to the effects of exposure to parabens, it’s necessary to provide the clear information concerning this relationship in the Table

-         please provide more detailed information he explanation is in Table 4 in comment to paper of Aker, 2016 (page 11), what dose or concentration of MP was associated with exactly 7.7% increase of SHBG?

Response: The author does not provide the information about dose or concentration in the cohort studied.

That’s not true, the explanation concerning the exposure is in the description of Table 3 of the citer paper- please read the paper once again and complete the information

D.    Tables – there are still some necessary corrections

-         please unify fonts in the Tables, according to the Template, unify the lines within Tables

-         in Tables 2-4 please consider to narrow the right and right column, and extend the columns in the middle, as the Tables are very long and have a lot of free space in first and last column

-         please leave one empty line below each Table  

-         please ensure that the titles of Tables are placed over particular Tables, not at the bottom of previous page (e.g. Table 2 should start below the Title of this Table, not on next page)

-         in the title of Table 3, benzylparaben should not be written with a capital letter, the same relates to ethylparaben in the title of Table 4

-         Tables are very long, please unify the lines and remove unnecessary empty lines

E.    Measurement units and missing data

-         please once more, confirm the units for PP concentrations: mg/L in line 13, while µg/L in line 217 (the second one are a thousand times lower than the first); please place appropriate references directly after cited values, as it is impossible to identify the source of particular data when six references is provided together for the whole paragraph (in line 217)

-         line 317 in 1st version – it is still unclear, if higher or lower levels of EP and PP were associated with reductions in tT4 and fT4 – please explain

F.    Figures

-         figures are very helpful, however they should be located close to the place where they are cited for the first time

-         the descriptions of Figures should be on the same page, not partially at the top of next page (e.g.  Fig. 1 is on page 15, a description on page 16)

G.    Please, clarify the sentence In Conclusion (page 18/19) “The HPT axis in humans and rodents has similarly, however the sensitivity of the axis to paraben exposure must vary”

H.    References

-         references are still not prepared according to the Instructions for Authors, provided by MDPI for the Journal, please prepare them accordingly; please unify writing the Titles of Journals with capital letters (e.g. Frontiers in Endocrinology, PLoS One, provide the year of publication in proper place

-         in the text, reference numbers should be placed in square brackets [ ], please unify listing all numbers of references or providing them as 37-42 (remove unnecessary coma), etc.

-         please verify, where are cited in the text References: 74, 90, 95, 100 (this one appears for the first time after reference 104), 106, 111-113 (these references are cited in the Tables but not in the main text, so they should not have so high numbers)?

Author Response

Reviewer 2  (round 2)

Comments and Suggestions for Authors

Thank you for resubmission a revised version of the paper „Environmental Endocrinology” and all explanations. Nevertheless, there are still some issues that require further corrections and the next round of review is necessary in my opinion.

  1. Graphical abstract – unfortunately, according to the Instructions for Authors, Graphical abstract should be an image, so, all text should be included within this image, e.g. abbreviations should be explained in the space of this image, without additional text that is definitely more expanded than just the legend – please correct accordingly;

Response: The abbreviations used in the graphical abstract were inserted in the figure. Thank you for the suggestion.

  1. Abbreviations

   -         please unify abbreviations in Graphical abstract, Abstract and and the main text (e.g. TH, HT, THs)

   -         abbreviation “HT”, used il line 9 of Abstract, is still not explained

   -         abbreviations: HPT, PPI, THs are explained over the Graphical abstract but not used in Figure;  instead, abbreviations TRH, IBP and TH are used in Figure but not explained – please correct more carefully

  -         please explain the abbreviation “SHGB” in Table 4, reference to Aker, 2019, if it relates to sex hormone binding globulin, it should be “SHBG” – not corrected after first review

-         please explain the abbreviation “PM” on page 7 in 5th line from the top

Response: The “PM” abbreviation was a formatting error that has now been corrected. Thank you for the observation. All other suggestions were accepted and corrected, thank you.

  1. Substantive remarks

-         I understand the answer of the Authors, however, the information that – despite searching in PubMed for publications since 1951 -the earliest cited publication on parabens is much younger – should also be placed in the main text to indicate, when really were published the papers included

Response: Thank you for the suggestion. We added this information in the main text.

-         in Table 2, the citation of paper of Aker, 2016 provides no information concerning the effects of buthylparaben, please re-write or remove; the same refers to Table 3, the citation of paper of Wang, 2015.

Response: In Table 2, the paper of Aker 2016 provides information concerning exposure to various parabens and other compounds (that are not the focus of this study). However, among the parabens tested, only butylparaben showed significant differences in the observed results, so this paper was added to this table. In Table 3, there was an error in the citation. The correct paper would be Aker's, 2016. We have corrected it in the table. Thank you for the suggestions.

New comment: if the paper provides information related to the effects of exposure to parabens, it’s necessary to provide the clear information concerning this relationship in the Table

Response: Thank you for the suggestion. We entered new information about the effects of BuP on the hormone levels, in table 2. 

-         please provide more detailed information he explanation is in Table 4 in comment to paper of Aker, 2016 (page 11), what dose or concentration of MP was associated with exactly 7.7% increase of SHBG?

Response: The author does not provide the information about dose or concentration in the cohort studied. 

That’s not true, the explanation concerning the exposure is in the description of Table 3 of the citer paper- please read the paper once again and complete the information.

Response: This information was added in table 4 (page 11). Thank you for your observation. 

  1. Tables – there are still some necessary corrections

-         please unify fonts in the Tables, according to the Template, unify the lines within Tables

-         in Tables 2-4 please consider to narrow the right and right column, and extend the columns in the middle, as the Tables are very long and have a lot of free space in first and last column

-         please leave one empty line below each Table  

-         please ensure that the titles of Tables are placed over particular Tables, not at the bottom of previous page (e.g. Table 2 should start below the Title of this Table, not on next page)

-         in the title of Table 3, benzylparaben should not be written with a capital letter, the same relates to ethylparaben in the title of Table 4

-         Tables are very long, please unify the lines and remove unnecessary empty lines

Response: Thank you for the suggestions. We have corrected all those things in the main text.

  1. Measurement units and missing data

-         please once more, confirm the units for PP concentrations: mg/L in line 13, while µg/L in line 217 (the second one are a thousand times lower than the first); please place appropriate references directly after cited values, as it is impossible to identify the source of particular data when six references is provided together for the whole paragraph (in line 217)

Response: The PP concentration units were adjusted and references were directly cited after the values cited in the paragraph. Thank you for the observations.

-         line 317 in 1st version – it is still unclear, if higher or lower levels of EP and PP were associated with reductions in tT4 and fT4 – please explain

Response: The increased levels of EP and PP were associated with reductions in T4 levels. We added this information in the main text. Thank you for this observation. 

  1. Figures

-         figures are very helpful, however they should be located close to the place where they are cited for the first time

-         the descriptions of Figures should be on the same page, not partially at the top of next page (e.g.  Fig. 1 is on page 15, a description on page 16)

  1. Please, clarify the sentence In Conclusion (page 18/19) “The HPT axis in humans and rodents has similarly, howeverthe sensitivity of the axis to paraben exposure must vary”

Response: Although the HPT axis functions similarly in rodents and humans, the sensitivity to paraben exposure can vary. In particular, if connections between human exposure to these chemicals and disturbance of the HPT axis are reported, the health effects on populations exposed to rodent thyroid-active compounds are of concern. Since the target organ, the thyroid gland, and the HPT axis have the same sites in rats and humans, the case for developing human chemical exposure standards based on rodent HPT axis disturbances is strong. This is particularly true given the lack of information on the toxicity of compounds that affect rodent thyroid function in humans, despite the fact that rodents and humans have significant quantitative physiological differences in the thyroid gland and the HPT axis. The use of rat toxicity studies for HPT axis disturbances to establish human health exposure standards for thyroid-active substances is significantly questionable as a result of these variations. 

  1. References

-         references are still not prepared according to the Instructions for Authors, provided by MDPI for the Journal, please prepare them accordingly; please unify writing the Titles of Journals with capital letters (e.g. Frontiers in Endocrinology, PLoS One, provide the year of publication in proper place

-         in the text, reference numbers should be placed in square brackets [ ], please unify listing all numbers of references or providing them as 37-42 (remove unnecessary coma), etc.

-         please verify, where are cited in the text References: 74, 90, 95, 100 (this one appears for the first time after reference 104), 106, 111-113 (these references are cited in the Tables but not in the main text, so they should not have so high numbers)? 

Response: The reference 100 was inserted incorrectly and we removed it from the text. In this way, the reference numbers between 100-117 were reorganized in the main text, the table and reference list. References 111-113 (cited only in the tables) became 110-112. Thank you for the suggestions, we have corrected them in the main text.